# Design and Experiment of a Portable Near-Infrared Spectroscopy Device for Convenient Prediction of Leaf Chlorophyll Content

**DOI:** 10.3390/s23208585

**Published:** 2023-10-19

**Authors:** Longjie Li, Junxian Guo, Qian Wang, Jun Wang, Ya Liu, Yong Shi

**Affiliations:** 1College of Mechanical and Electrical Engineering, Xinjiang Agricultural University, Urumqi 830052, China; lidelongjie@163.com (L.L.); hebau_wangqian@163.com (Q.W.); 320232352@xjau.edu.cn (J.W.); zztyly@163.com (Y.L.); shiyong19860324@163.com (Y.S.); 2Key Laboratory of Xinjiang Intelligent Agricultural Equipment, Urumqi 830052, China

**Keywords:** Internet of Things technology, spectral data analysis, chlorophyll content prediction, regression algorithms, data preprocessing

## Abstract

This study designs a spectrum data collection device and system based on the Internet of Things technology, aiming to solve the tedious process of chlorophyll collection and provide a more convenient and accurate method for predicting chlorophyll content. The device has the advantages of integrated design, portability, ease of operation, low power consumption, low cost, and low maintenance requirements, making it suitable for outdoor spectrum data collection and analysis in fields such as agriculture, environment, and geology. The core processor of the device uses the ESP8266-12F microcontroller to collect spectrum data by communicating with the spectrum sensor. The spectrum sensor used is the AS7341 model, but its limited number of spectral acquisition channels and low resolution may limit the exploration and analysis of spectral data. To verify the performance of the device and system, this experiment collected spectral data of Hami melon leaf samples and combined it with a chlorophyll meter for related measurements and analysis. In the experiment, twelve regression algorithms were tested, including linear regression, decision tree, and support vector regression. The results showed that in the original spectral data, the ETR method had the best prediction effect at a wavelength of 515 nm. In the training set, RMSE_c_ was 0.3429, and R_c_^2^ was 0.9905. In the prediction set, RMSE_p_ was 1.5670, and R_p_^2^ was 0.8035. In addition, eight preprocessing methods were used to denoise the original data, but the improvement in prediction accuracy was not significant. To further improve the accuracy of data analysis, principal component analysis and isolation forest algorithm were used to detect and remove outliers in the spectral data. After removing the outliers, the RFR model performed best in predicting all wavelength combinations of denoised spectral data using PBOR. In the training set, RMSE_c_ was 0.8721, and R_c_^2^ was 0.9429. In the prediction set, RMSE_p_ was 1.1810, and R_p_^2^ was 0.8683.

## 1. Introduction

With the development and popularization of Internet of Things (IoT) technology, the application scope of sensors and devices is becoming increasingly wide [1,2,3,4]. Spectrum data can reflect the characteristics and properties of substances [5], while providing object recognition and classification information [6]. In the field of agriculture, the collection and analysis of spectrum data are of great significance for nutrient management and growth monitoring of Hami melon crops [7]. Hami melon is an important economic crop [8], and photosynthesis is crucial for ensuring healthy growth and high yield of plants [9]. Chlorophyll, as an important photosynthetic pigment in plants [10], is closely related to the photosynthetic rate and growth status of plants [11,12]. By measuring and analyzing spectrum data of plants, critical information such as chlorophyll content can be obtained non-destructively [13]. Jeremy Aditya Prananto first evaluated the ability of handheld near-infrared spectrometers to predict nutrient content in dried and ground cotton leaf samples, demonstrating that handheld near-infrared spectrometers are a practical choice for accurately measuring leaf nutrient concentration [14]. Lihua Liu found that near-infrared spectroscopy can be used as an alternative method for real-time quantitative and monitoring of chlorophyll during the processing of Tiancha [15], providing scientific guidance for agricultural production. Jianfeng Zhang developed a new method for estimating winter wheat leaf chlorophyll content based on visible and near-infrared sensors [16]. Yu-Jie Wang used a miniature near-infrared spectrometer to evaluate the pigment content of two field tea trees. A miniature near-infrared system based on a smartphone can quickly, non-destructively, and inexpensively diagnose plant nutrition status [17]. Xiu Jin used a handheld miniature near-infrared spectrometer to analyze the nitrogen, phosphorus, and potassium content of nutrient-deficient pear leaf samples, with this method able to quickly predict nutrient deficiency during the cultivation period of pear leaves [18], promoting crop quality and yield improvement [19]. However, traditional spectral data collection and analysis processes have some shortcomings. For example, traditional spectral data collection methods require the use of complex fiber optic cables to transmit spectral information and rely on heat-generating light source equipment and spectral analyzer equipment to collect data with a computer. The large size of the entire device is inconvenient for outdoor collection operations, and the user interface is simple and cannot meet the requirements of data visualization and real-time interaction.

To solve the above problems, this study proposes a spectrum data collection device and system based on IoT technology (see Appendix A for details). Through the combination of cloud servers, collection devices, and interactive interfaces, the system achieves automatic collection, storage, and data visualization of spectral data. The cloud server deploys a series of necessary software services using Docker containerization technology to achieve efficient data reception, secure storage, and flexible interaction. The collection device integrates advanced microcontrollers and spectral sensors to achieve accurate collection and transmission of spectral data. The interactive interface uses Websocket technology to achieve real-time synchronization of front-end and back-end data and user visualization operations.

This study demonstrates non-destructive detection of chlorophyll content in muskmelon leaves. By collecting spectral data and chlorophyll content measurements of muskmelon plants at different growth stages and nutrient states, a prediction model can be established, and chlorophyll content can be predicted by analyzing the spectral data of new samples [20]. Such a prediction model provides important reference for agricultural production, helping farmers and agricultural experts adjust fertilization and nutrient management strategies in a timely manner [21], and maximizing the yield and quality of muskmelon [22]. The experiment proves that the spectral data collected using IoT technology can accurately predict the chlorophyll content of leaves through various data processing and analysis. At the same time, the design of the interactive interface enables users to conveniently operate and explore spectral data and obtain real-time collection results. Therefore, the research results have important theoretical significance and practical application value, providing a new solution for the collection and analysis of spectral data, and providing strong support for practical application in agriculture, environmental monitoring, and other related fields.

## 2. Overall Design of the Device

As shown in Figure 1, the collection device ensures that equidistant spectral data can be obtained every time through the spectral sensor (6) and leaf fixing plate (7). The operation of the device mainly depends on the cloud server, collection device, and interactive interface. The cloud server adopts Docker containerization technology to deploy the EMQX, Node-RED, InfluxDB, and Flask environments, realizing the reception, storage, and data interaction of the data input by the web page users and the spectral data collected by the collection device. The collection device uses the ESP8266-12F microcontroller (hereinafter referred to as the microcontroller) as the core processor, communicates with the spectral sensor to obtain spectral data, and transmits the collected data to the cloud server through WiFi wireless network. The interactive interface uses Websocket technology to achieve real-time synchronization of front-end and back-end data, and visualizes the data through chart functions, providing intuitive data analysis and display functions.

### 2.1. Deployment of IoT Server

As shown in Figure 2, the server is configured with one core CPU, 2 GB memory, 40 GB system disk, and 1 Mbps public network bandwidth, running on Ubuntu 16.04.6 LTS x86_64. EMQX is responsible for handling the access to MQTT communication devices and forwarding topic data to ensure efficient data transmission. Node-RED serves as a flow orchestration tool, listening to and capturing the spectrum data collected by the AS7341 spectrum device and the data input by the user on the web page in real-time and forwarding them to the InfluxDB database for reliable data storage. Meanwhile, Flask is used to subscribe to real-time spectrum data in the MQTT server and the data saved in the final database and transmit said data to the front-end for visualization display through WebSocket.

### 2.2. Spectral Acquisition Device

The spectral acquisition device, as shown in Figure 3, uses a two-layer PCB circuit board without copper for testing. The device is powered by a 5 V power bank, and the spectral sensor used is the AS7341, with a data acquisition channel covering a range of 415 nm to 940 nm. The collected spectral data are efficiently processed by the microcontroller through I^2^C communication. As shown in Figure 4, the microcontroller packages the sensor data and wirelessly transmits it to the cloud server via a shared network WiFi connection with a mobile phone (see Appendix B for details). Users can view the real-time visualization of the latest collected spectral data through a web interface. At the same time, the microcontroller receives information on light intensity, acquisition times, and data calibration instructions from the server. By using the light intensity information to control the current of the LED, the brightness of the LED can be adjusted. Meanwhile, by setting the original spectral acquisition times based on the acquisition times information, the average value calculation is performed to improve the reliability and accuracy of the data. By receiving spectral calibration instructions, the data correction and compensation parameters are used to adjust the whiteboard spectral curve to the same level, ensuring the accuracy and consistency of the data.

### 2.3. Interactive Interface Design

The collection interface is shown in Figure 5. The back-end uses Flask to establish a communication connection with the MQTT server, subscribes to the spectral data sent by the spectral acquisition device, and uses WebSocket to transmit the spectral data to the front-end for real-time rendering and display. The front-end is responsible for receiving and rendering the spectral data sent by the back-end and for providing users with the function to input experimental sample numbers, leaf temperature, and chlorophyll data. The front-end sends the user-input data to the back-end through an interface. After receiving the data submitted by the user through the web page, the back-end splices it with the latest spectral data, packages it, and sends it to the MQTT server for data forwarding and storage in the InfluxDB database. The latest database data are visualized and displayed in the front-end, allowing operators to confirm whether their submitted data have been successfully saved. In addition, the functionality provided by InfluxDB allows for the download of saved data, including combined data of user-submitted data and spectral data, reducing the workload of experimental personnel in secondary data statistics, and providing an efficient and reliable data processing and analysis environment for them.

### 2.4. Data Collection 

The experiment was conducted at the experimental field of Xinjiang Academy of Agricultural Sciences, with longitude of 87.476325 and latitude of 43.949915. A Top Cloud-agri TYS-4N chlorophyll meter was selected, with a measurement range of 0.0–99.9 SPAD and an accuracy of ±1.0 SPAD. The SPAD values of the leaves of Hami melon were measured, avoiding measuring over the thick veins to ensure the accuracy of the measurement results [23]. The SPAD values and spectral data of 100 different plant leaf samples were measured outdoors, and the collected spectral data are shown in Figure 6a. Before collecting spectral data, the hardware acquisition parameters need to be set in the web setting area of the collection device. In the “Acquisition Times” position on the web page, the average collection times should be set to 3, and the LED current size should also be set to 3 in the “LED” position. After completing these parameter settings, click the “Setting” button to submit the set parameters. During the subsequent spectral data collection, three original average spectral data points of the leaf samples under fixed light intensity can be obtained [24].

## 3. Results

### 3.1. Prediction of Raw Spectral Data 

The original spectral data are shown in Figure 6a. The entire dataset was randomly divided into a 70:30 ratio for model training and prediction, respectively. Twelve regression algorithms were used to analyze the spectral data and SPAD values, including linear regression (LR) [25], K-nearest neighbor regression (KNN) [26], support vector regression (SVR) [27], ridge regression (RR) [28], Lasso regression (Lasso) [29], decision tree regression (DTR) [30], extremely randomized tree regression (ETR) [31], random forest regression (RFR) [32], AdaBoost regression (ABR) [33], gradient boosting regression (GBR) [34], bagging regression (BAR) [35], and partial least squares regression (PLSR) [36] (see Appendix C for details). For each collection band and all band combinations between 415 nm and 940 nm, the twelve regression algorithms were used for analysis and prediction. The best prediction results for each regression analysis are shown in Table 1. It was found that ETR, RFR, and BAR performed well in predicting SPAD values at a wavelength of 515 nm, with ETR attaining the best prediction performance. On the training set, the RMSE_c_ of ETR was 0.3429 and R_c_^2^ was 0.9905. On the prediction set, the RMSE_p_ of ETR was 1.5670 and R_p_^2^ was 0.8035. The model prediction performance is shown in Figure 9a.

### 3.2. Spectral Data Denoising Analysis

Considering the noise interference from environmental light during the collection of original spectral data from leaf samples [37], eight widely used preprocessing methods were employed to eliminate the effects of scattering and noise on the original data. These methods include Multiplicative Scatter Correction (MSC) [38], Standard Normal Variate (SNV) [39], Discrete Wavelet Transform (DWT) [40], Savitzky-Golay (SG) smoothing [41], MinMax scaling (MinMax) [42], Outlier Detection (OD) [43], Percentile-based outlier removal (PBOR) [44], and Continuum Removal (CR) [45]. The preprocessed spectral curves are shown in Figure 7. To evaluate the effects of these eight denoising preprocessing methods on prediction accuracy, ETR, RFR, and BAR, the three regression algorithms with the highest prediction accuracy in the original spectra, were used to perform regression analysis on each denoised spectrum. The analysis results are shown in Table 2. By comparing the regression results of each denoising method with the RMSE_c_, R_c_^2^, RMSE_p_, and R_p_^2^ values in Table 1, it was found that DWT and OD showed a trend of decreasing prediction accuracy, while the other preprocessing methods did not significantly improve the prediction accuracy.

### 3.3. Data Dimensionality Reduction and Outlier Removal

To eliminate the interference of abnormal samples in the denoised and original spectral data, Principal Component Analysis (PCA) was used to reduce the dimensionality of the data [46], with a final dimension of three and projection onto a three-dimensional space. An Isolation Forest (IF) algorithm was used to detect and distinguish abnormal values based on the distribution of each sample point in the three-dimensional space [47]. The classification results of the original spectral data and the denoised spectral data are shown in Figure 6b and Figure 8, respectively. After removing the abnormal values from the original and denoised spectral data, ETR, RFR, and BAR regression analysis were performed separately. The specific analysis results are shown in Table 3, and the judgment analysis was compared with the RMSE_c_, R_c_^2^, RMSE_p_, and R_p_^2^ of Table 1. The prediction accuracy of ETR decreased significantly, while in RFR and BAR, the prediction accuracy increased significantly. Among them, the effect of RFR was the best. Compared with the highest precision model without removing the abnormal values, the RMSE_p_ decreased from 1.5670 to 1.3456, and the R_p_^2^ increased from 0.8035 to 0.8358. After removing the abnormal values from the denoised spectral data, all the denoising data prediction accuracies improved, and RFR showed the best accuracy in predicting all bands under PBOR denoising. Compared with the highest precision model of the original and denoised spectral data, the RMSE_p_ decreased from 1.5670 to 1.1810, and the R_p_^2^ increased from 0.8035 to 0.8683. Therefore, RFR showed the best modeling effect and stability in predicting chlorophyll in all bands under PBOR denoising. The model predictions are shown in Figure 9b.

## 4. Discussion

This study successfully designed a spectrum data acquisition device and system based on the IoT technology, which integrated the AS7341 spectral sensor and ESP8266 microcontroller to achieve integrated spectrum data acquisition. Compared with traditional spectrometers, the spectrum acquisition device designed in this paper has many advantages and some disadvantages.

Advantages: integrated design. The ESP8266 microcontroller, AS7341 spectral sensor, and LED light source are integrated, which is convenient for outdoor spectrum data collection. At the same time, since there is no exposed signal transmission part, the impact of environmental light is relatively reduced. Portability: it has a small size and weight which render it convenient to carry and use. In addition, there is no need to pay attention to the bending of the optical fiber during the collection process, thus removing the risk of breakage. Easy to operate and low power consumption: the microcontroller, AS7341 spectral sensor, and LED light source have low power consumption, solving the problem of outdoor long-term power supply. This integrated design makes spectrum acquisition more portable and easier to operate and is suitable for spectrum data collection and analysis in outdoor environments, as it is often required for, e.g., agricultural, environmental, and geological purpose. Low cost and maintenance requirements: it has the characteristics of low cost and easy maintenance, which is suitable for some applications with high requirements for cost and maintenance, such as scientific and teaching experiments.

Disadvantages: fewer channels. The AS7341 spectral sensor can collect data from up to 14 channels, while traditional spectrometers can collect data from hundreds of channels. Therefore, the effective resolution of AS7341 data is low, and many effective information may be missed during the collection process, which limits the exploration of spectral data. However, the 14 channels collected are sufficient to support most spectral analysis applications.

In this study, we successfully designed a spectrum data acquisition device and system based on the Internet of Things (IoT) technology. The collection and processing of spectral data were achieved through integrated design and the use of AS7341 spectral sensors and ESP8266 microcontrollers. Our research results show that the device has many advantages, such as portability, ease of operation, and low power consumption, but also some limitations, such as limited channel numbers and low resolution. In future research, we will explore ways to further optimize the performance of the device by using transmission or reflection combined with transmission, in order to meet the demand for higher-precision spectral analysis. At the same time, we will also attempt to improve the light source part to increase the device’s collection efficiency and accuracy. In terms of power supply, we will use a battery and device integration design to make data collection more portable. These improvements will help enhance the performance and reliability of the device, thereby providing more accurate and reliable data support for related research.

## 5. Conclusions

Based on the designed spectral acquisition system and equipment, the spectral data acquisition and SPAD regression prediction analysis of Hami melon leaf samples were completed. The regression analysis results showed that at the wavelength of 515 nm, ETR, RFR, and BAR regression algorithms could all predict SPAD well, among which ETR attained the best prediction effect. On the training set, RMSE_c_ was 0.3429, and R_c_^2^ was 0.9905; on the prediction set, RMSE_p_ was 1.5670, and R_p_^2^ was 0.8035. After denoising the spectral data, the improvement in prediction accuracy was not significant. Through the use of principal component analysis and Isolation Forest algorithm, abnormal points were successfully detected and removed. After removing the outliers, the prediction accuracy of the dataset, as well as the raw and denoised spectral data, were improved. Among all wavelength combination for spectral data predictions after PBOR denoising, the RFR model performed best. On the training set, RMSE_c_ was 0.8721 and R_c_^2^ was 0.9429; on the prediction set, RMSE_p_ was 1.1810 and R_p_^2^ was 0.8683. This piece of acquisition equipment offers several advantages, including small size, convenient portability, cloud access, and real-time visualization of spectral data, providing technical reference value for the development and application of intelligent spectral devices in agriculture.

## Figures and Tables

**Figure 1 sensors-23-08585-f001:**
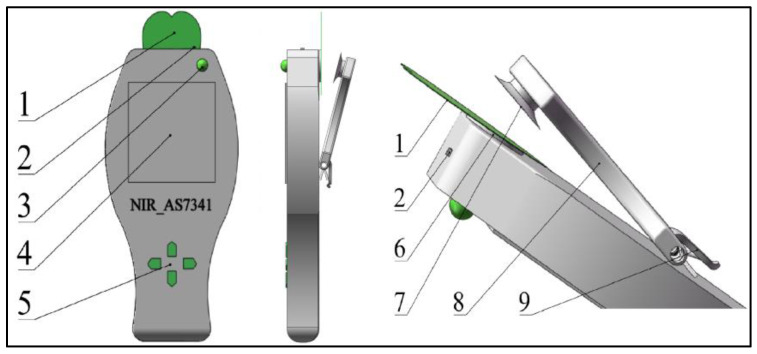
Structure of the Acquisition Device. 1. Leaf blade; 2. power switch; 3. indicator light; 4. display screen; 5. function keys; 6. AS7341; 7. fixing plate; 8. force arm; 9. switch.

**Figure 2 sensors-23-08585-f002:**
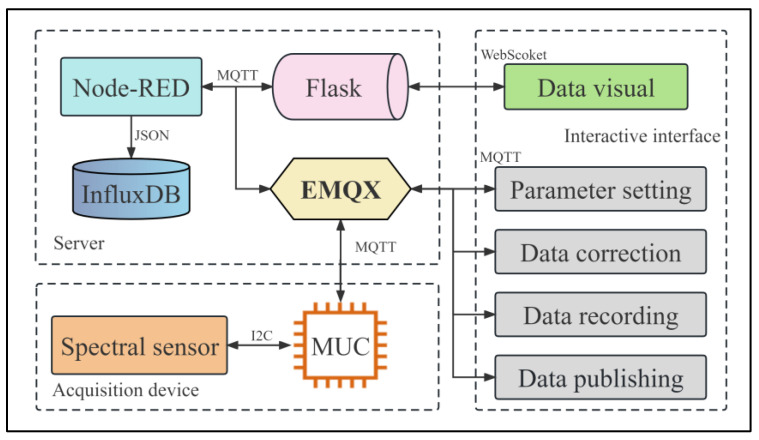
Overall functional architecture.

**Figure 3 sensors-23-08585-f003:**
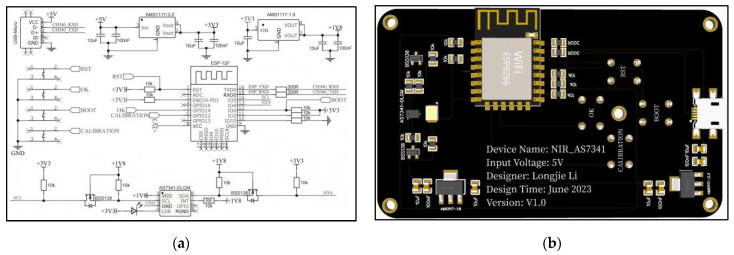
Spectral acquisition device. (**a**) Schematic diagram; (**b**) PCB diagram.

**Figure 4 sensors-23-08585-f004:**
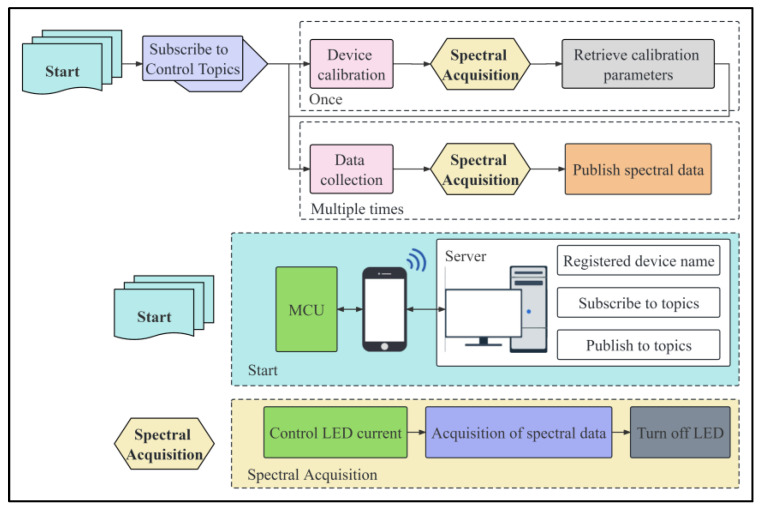
Spectral acquisition flowchart.

**Figure 5 sensors-23-08585-f005:**
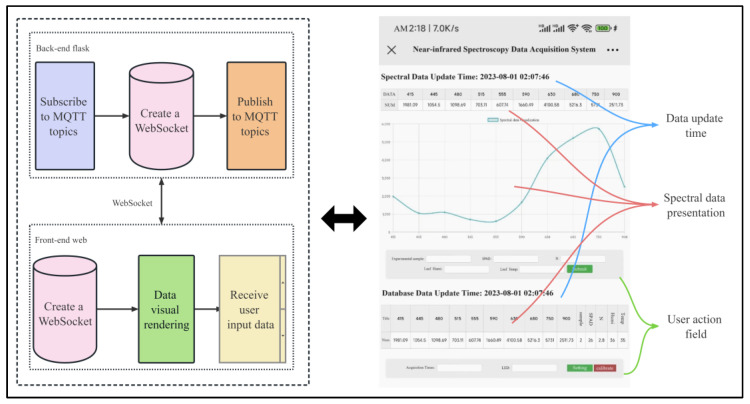
Interactive interface design.

**Figure 6 sensors-23-08585-f006:**
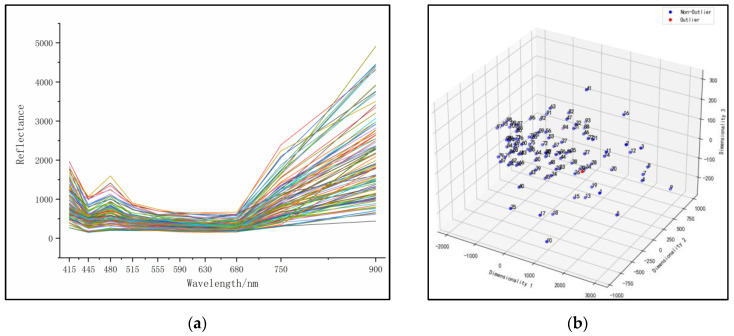
Raw spectral data. (**a**) Data curves; (**b**) anomaly removal.

**Figure 7 sensors-23-08585-f007:**
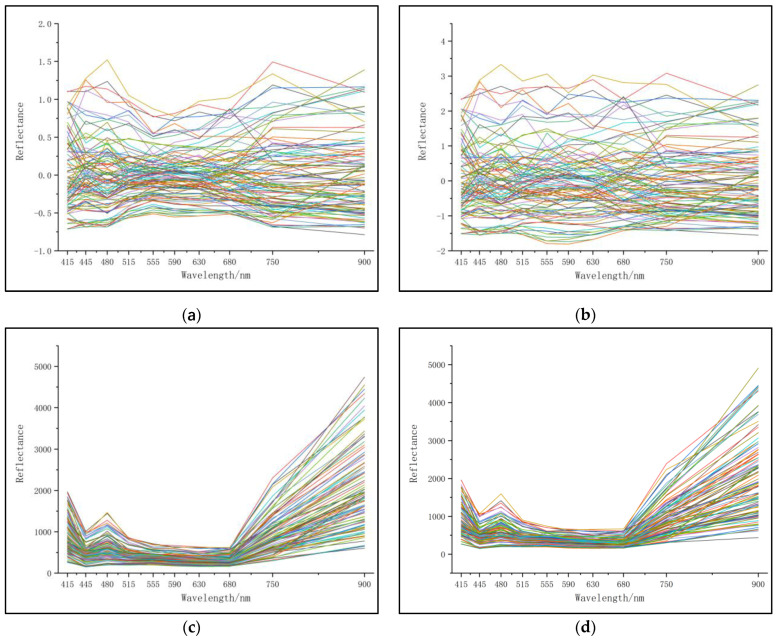
Spectral data noise reduction processing. (**a**) MSC; (**b**) SNV; (**c**) DWT; (**d**) SG; (**e**) MinMax; (**f**) OD; (**g**) PBOR; (**h**) CR.

**Figure 8 sensors-23-08585-f008:**
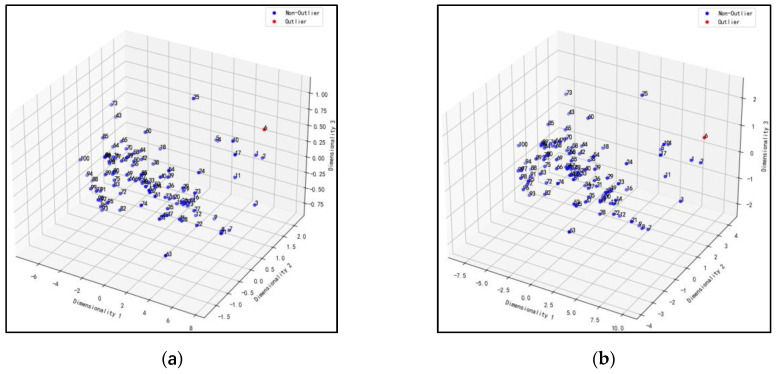
Abnormal value elimination distribution map. (**a**) MSC; (**b**) SNV; (**c**) DWT; (**d**) SG; (**e**) MinMax; (**f**) OD; (**g**) PBOR; (**h**) CR.

**Figure 9 sensors-23-08585-f009:**
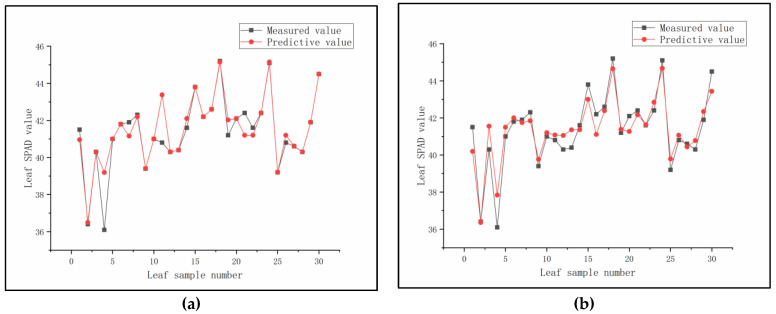
SPAD prediction model. (**a**) Prediction of raw spectra at 515 nm using ETR regression model; (**b**) Prediction of de-noised spectra in all bands of PBOR using RFR regression model.

**Table 1 sensors-23-08585-t001:** Original spectrum and SPAD analysis.

Forecasting Method	Wavelength (nm)	Training Set		Prediction Set	
		RMSE_c_	R_c_^2^	RMSE_p_	R_p_^2^
LR	590	1.9273	0.6987	2.0030	0.6691
KNN	515	1.5182	0.8130	1.6608	0.7792
SVR	515	1.8824	0.7126	1.8471	0.7269
RR	590	1.9273	0.6987	2.0330	0.6691
Lasso	590	1.9273	0.6987	2.0330	0.6691
DTR	515	0.3429	0.9905	1.7449	0.7563
**ETR**	**515**	**0.3429**	**0.9905**	**1.5670**	**0.8035**
RFR	515	0.7062	0.9595	1.5798	0.8002
ABR	515	1.1206	0.8981	1.6480	0.7826
GBR	515	0.4692	0.9821	1.6986	0.7690
BAR	515	0.8010	0.9480	1.5777	0.8008
PLSR	590	1.9273	0.6987	2.0330	0.6691

**Table 2 sensors-23-08585-t002:** Noise reduction analysis of spectral data.

Pretreatment Method	Forecasting Method	Wavelength (nm)	Training Set		Prediction Set	
			RMSE_c_	R_c_^2^	RMSE_p_	R_p_^2^
**MSC**	**ETR**	**515**	**0.3429**	**0.9905**	**1.5670**	**0.8035**
	RFR		0.7061	0.9596	1.5826	0.7995
	BAR		0.8010	0.9480	1.5777	0.8008
**SNV**	**ETR**	**515**	**0.3429**	**0.9905**	**1.5670**	**0.8035**
	RFR		0.7061	0.9596	1.5826	0.7995
	BAR		0.8010	0.9480	1.5777	0.8008
DWT	ETR	555 590 555	0.0720	0.9996	2.3676	0.5513
	RFR		0.7567	0.9536	2.2347	0.6002
	BAR		1.0597	0.9089	2.2184	0.6061
**SG**	**ETR**	**515**	**0.3429**	**0.9905**	**1.5670**	**0.8035**
	RFR		0.7062	0.9595	1.5798	0.8002
	BAR		0.8010	0.9480	1.5777	0.8008
**MinMax**	**ETR**	**515**	**0.3429**	**0.9905**	**1.5670**	**0.8035**
	RFR		0.7044	0.9598	1.5798	0.8002
	BAR		0.7989	0.9482	1.5777	0.8008
OD	ETR	590	0.1737	0.9971	1.8834	0.6074
	RFR		0.7179	0.9508	1.7394	0.6651
	BAR		0.8133	0.9369	1.7986	0.6419
PBOR	ETR	515	0.3618	0.9894	1.5813	0.7998
	RFR		0.7106	0.9590	1.5853	0.7988
	BAR		0.8079	0.9471	1.5679	0.8032
**CR**	**ETR**	**515**	**0.3429**	**0.9905**	**1.5670**	**0.8035**
	RFR		0.7061	0.9596	1.5826	0.7995
	BAR		0.8010	0.9480	1.5777	0.8008

**Table 3 sensors-23-08585-t003:** Outlier rejection analysis.

Pretreatment Method	Pretreatment Method	Forecasting Method	Wavelength (nm)	Training Set		Prediction Set	
				RMSE_c_	R_c_^2^	RMSE_p_	R_p_^2^
Non	1	ETR	All	0.1665	0.9978	1.6408	0.7558
		RFR		0.8736	0.9394	1.3456	0.8358
		BAR		1.1249	0.8996	1.3799	0.8273
MSC	1	ETR	515 All	0.1665	0.9978	1.6408	0.7558
		RFR		0.8736	0.9394	1.3467	0.8355
		BAR		1.1249	0.8996	1.3858	0.8258
SNV	1	ETR	515 All	0.1665	0.9978	1.6408	0.7558
		RFR		0.8736	0.9394	1.3456	0.8358
		BAR		1.1249	0.8996	1.3799	0.8273
DWT	1	ETR	590	0.0904	0.9993	2.1442	0.6252
		RFR		0.9712	0.9222	1.8656	0.7162
		BAR		1.1681	0.8874	1.9453	0.6915
SG	1	ETR	515	0.1665	0.9978	1.6408	0.7558
		RFR		0.8736	0.9394	1.3456	0.8358
		BAR		1.1249	0.8996	1.3799	0.8273
MinMax	1	ETR	515	0.1665	0.9978	1.6408	0.7558
		RFR		0.8736	0.9394	1.3461	0.8356
		BAR		1.1249	0.8996	1.3858	0.8258
OD	1	ETR	515 All 555	0.5161	0.9719	1.9488	0.6732
		RFR		0.7140	0.9462	1.7327	0.7417
		BAR		0.9447	0.9059	1.9373	0.6771
**PBOR**	**1**	ETR	555 **All** 515	0.4962	0.9815	1.4004	0.8148
		**RFR**		**0.8721**	**0.9429**	**1.1810**	**0.8683**
		BAR		1.0745	0.9133	1.4931	0.7895
CR	1	ETR	515 All All	0.1665	0.9978	1.6408	0.7558
		RFR		0.8736	0.9394	1.3467	0.8355
		BAR		1.1249	0.8996	1.3858	0.8258

## Data Availability

Data available on request from the authors.

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
