# Peer review of "Design and Experiment of a Portable Near-Infrared Spectroscopy Device for Convenient Prediction of Leaf Chlorophyll Content"

_sensors, 2023, doi:10.3390/s23208585_

Round 1

Reviewer 1 Report

This paper presents a Leaf Chlorophyll Content detection device built on the foundation of a low-cost AS7341 spectral color sensor, a wireless microcontroller, and a machine learning system to predict Chlorophyll Content from the spectral signal. The content is interesting and the topic is of clear relevance. Nevertheless, several issues with the current manuscript need attention before it is suitable for publication.

There is noticeable repetition in certain sections, which overlap with others. I recommend revisiting the structure to minimize such redundancies.

The abstract contains an abundance of details. Currently, it begins with descriptions of the services and hardware, pivots to experimental results, and then revisits the hardware, including its pros and cons. A more logical flow would be to conclude discussions on the hardware (highlighting its strengths and weaknesses) before delving into the experiments. Furthermore, it would be prudent to introduce the experiments slightly earlier, leading into the results, e.g., "Twelve regresion algorithms were tested and the results show that..."

Regarding the device's design, could you provide more insight into the hardware? Is this a prototype or designed for mass production? Does it incorporate a custom PCB or standalone modules? Incorporating actual images of the prototype alongside descriptions of its components would enrich the paper. If this is not feasible, kindly elucidate why.

The description of Figure 3 could be a little more detailed, specially for the flow chart.

Figure 5 caption is wrong.

Line 169: "the parameter of collecting original average spectral data of leaf samples was submited on the web page..." I cannot understand what this sentence want to state, it probably should be rewritten or needs further clarification.

In Table 1, using the acronym "LR" to represent two distinct algorithms could be confusing. A differentiation in their nomenclature would be helpful.

Line 222, "the RMSE increased from..." have structural issues that make it hard to understand. Appart from this one there are some other hard to read sentences along the manuscript. I recommend a thorough grammatical and structural review.

Figure 8 caption needs more information about what represents figure a versus figure b.

Tables 2 and 3 share identical captions. Kindly provide distinct and descriptive captions for each.

The data in Tables 2 and 3 is not adequately introduced in the main text. A comprehensive introduction is necessary. Additionally, Table 3 has a redundant column titled "Pretreatment method."

In the discussion, you mention future optimizations of the device's performance. Could you specify the areas of improvement?

Regarding the device itself, it is stated that WiFi technology is used to transfer data in real time. Why was this tecnology used? What provides the network in the field? Are there no connection issues taking into account it is a device meant to be used in the field? Have any empirical tests been performed regarding connectivity? Have other technologies been taken into account?

Lastly, I have reservations about Appendix A. The detailed description of the sensor might be superfluous for the content of this paper, especially when its datasheet is readily available online.

The overall quality of the manuscript is good but there are some sentences that have some error or the construction make it hard to understand. Some of them were already pointed out along the review, but a thorough grammatical and structural review is recommended.

Author Response

1There is noticeable repetition in certain sections, which overlap with others. I recommend revisiting the structure to minimize such redundancies.

Response:Thank you very much reviewer for your comments, I have now reduced this duplication after extensive reorganization of the full text.

2The abstract contains an abundance of details. Currently, it begins with descriptions of the services and hardware, pivots to experimental results, and then revisits the hardware, including its pros and cons. A more logical flow would be to conclude discussions on the hardware (highlighting its strengths and weaknesses) before delving into the experiments. Furthermore, it would be prudent to introduce the experiments slightly earlier, leading into the results, e.g., "Twelve regresion algorithms were tested and the results show that..."

Response:Thank you very much for your comments, reviewer, I have benefited a lot here, I have carefully followed the completion of the comments you gave to complete the appropriate changes.

3Regarding the device's design, could you provide more insight into the hardware? Is this a prototype or designed for mass production? Does it incorporate a custom PCB or standalone modules? Incorporating actual images of the prototype alongside descriptions of its components would enrich the paper. If this is not feasible, kindly elucidate why.

Response:Thank you very much for your comments, reviewer. The PCB circuit board I used is a two-layer circuit board with an uncoated copper design, which is intended to avoid interfering with the WiFi signal of the ESP8266-12F. This is a prototype design that has not yet been mass-produced. I have attached the corresponding PCB circuit board image at position 3(b) in Figure 3.

4The description of Figure 3 could be a little more detailed, specially for the flow chart.

Response:Thank you very much for your comments, reviewer. I have moved the original Figure 3 to the position of Figure 4, and made detailed revisions as requested by you.

5Figure 5 caption is wrong.

Response:Thank you very much for your comments, reviewer. I have corrected the erroneous figure caption as per your request.

6Line 169: "the parameter of collecting original average spectral data of leaf samples was submited on the web page..." I cannot understand what this sentence want to state, it probably should be rewritten or needs further clarification.

Response:Thank you very much for your comments, reviewer. I have rephrased the sentence as follows: Prior to collecting spectral data, it is necessary to set the hardware acquisition parameters in the settings area of the collection device's web page. Set the acquisition averaging to 3 at the "Acquisition Times" location on the web page, and set the LED light current to 3 at the "LED" location. After completing these parameter settings, click the "Setting" button to submit the settings. During subsequent spectral data collection, three sets of raw average spectral data of the leaf sample under fixed light intensity can be obtained.

7In Table 1, using the acronym "LR" to represent two distinct algorithms could be confusing. A differentiation in their nomenclature would be helpful.

Response:Thank you very much for your comments, reviewer. I have followed your suggestion and changed the abbreviation of "LR" to "Lasso" in reference to Lasso regression.

8Line 222, "the RMSE increased from..." have structural issues that make it hard to understand. Appart from this one there are some other hard to read sentences along the manuscript. I recommend a thorough grammatical and structural review.

Response:Thank you very much for your comments, reviewer. I have thoroughly reviewed and rewritten this section according to your suggestions.

9Figure 8 caption needs more information about what represents figure a versus figure b.

Response:Thank you very much for your comments, reviewer. I have moved the original Figure 8 to the position of Figure 9 and added the required explanation, which reads as follows: (a) original spectra prediction of the 515nm band using the ETR regression model, and (b) denoised spectra prediction of all bands under PBOR using the RFR regression model.

10Tables 2 and 3 share identical captions. Kindly provide distinct and descriptive captions for each.

Response:Thank you very much for your comments, reviewer. I have corrected the table caption as per your request.

11The data in Tables 2 and 3 is not adequately introduced in the main text. A comprehensive introduction is necessary. Additionally, Table 3 has a redundant column titled "Pretreatment method."

Response:Thank you very much for your comments, reviewer. I have made the corresponding modifications as requested:

  1. Further introduction of the data in Tables 2 and 3 was provided. By comparing the regression analysis before and after denoising and outlier removal, changes in RMSEc, Rc2, RMSEp, and Rp2 were determined to evaluate the effectiveness of the prediction.
  2. In addition, due to the lack of clarity in the previous paper, the column "preprocessing method" in Table 3 is redundant. In fact, the column of "preprocessing method" is necessary because it can indicate the regression prediction analysis after removing outliers under the original and preprocessed spectral conditions, and further determine the effect of outlier removal on the prediction accuracy of the regression analysis in a certain noise reduction situation and under a certain regression analysis condition.

12In the discussion, you mention future optimizations of the device's performance. Could you specify the areas of improvement?

Response:Thank you very much for your comments, reviewer. I have supplemented the content regarding the optimization of device performance as follows: In future studies, we will explore the use of transmission or reflection combined with transmission to further optimize the performance of the device, in order to meet the higher accuracy requirements of spectral analysis. At the same time, we will also attempt to improve the light source to enhance the device's data acquisition efficiency and accuracy. In terms of power supply, we will use battery and device integration design to make data collection more portable. These improvement measures will help to enhance the performance and reliability of the device, providing more accurate and reliable data support for related research.

13Regarding the device itself, it is stated that WiFi technology is used to transfer data in real time. Why was this tecnology used? What provides the network in the field? Are there no connection issues taking into account it is a device meant to be used in the field? Have any empirical tests been performed regarding connectivity? Have other technologies been taken into account?

Response:Thank you for your comment, reviewer. We chose to use WiFi for data transmission for two reasons. Firstly, it is more convenient to directly connect to the WiFi hotspot shared by the smartphone, as we also need to use the smartphone to submit and view data during the data collection process. This way, we can use the smartphone's shared network hotspot to provide network access for the device. Secondly, using this network form allows us to directly transmit the spectral data we collected to a cloud server, so that we can remotely view and download the data. There were no connection issues with this method, and we also conducted relevant data communication tests on this connection method, which are detailed in Appendix C.

14Lastly, I have reservations about Appendix A. The detailed description of the sensor might be superfluous for the content of this paper, especially when its datasheet is readily available online.

Response:Thank you very much for your comments, reviewer. I have deleted the original Appendix A.

Reviewer 2 Report

This paper introduces a spectral data collection device and system based on Internet of Things technology, aimed at solving the cumbersome steps in collecting leaf chlorophyll and providing a more convenient and accurate method for predicting leaf chlorophyll content.

 I have several comments for authors to improve their paper as follows:

1. The novelty of the paper is not justified.

2. The figure of the paper is not clear and up to the mark.

2. It is desired to discuss the model complexity of the proposed model and the baseline models. This would show their efficiency in terms of runtime and space complexity.

3. I suggest the authors give a problem statement or use case in Methodology so as to help readers understand how to make a recommendation based on the dataset.

4. I was somewhat confused about the motivations described in the introduction. I’m not sure if the first paragraph is fully necessary since it frames the problem as reducing detection/response time after a fall in order to get a response; is that the primary goal or is it to detect the fall prior to occurrence and prevent it (at no point do the results compare timing for estimating a pre-fall or fall prior to occurrence)? What are the present timescales for detecting and alerting to a fall and how much matters? For example, if a fall has already happened, does a difference between a 0.2s and 2s alert actually matter when the response time may be substantially greater? This would be useful to know whether the detections actually happen in time.

5. Do you anticipate using the generated model as is on new data from people who the model has not been trained on or are you expecting to need to expand the training set and include each individual who uses such a device in the training set?

Author Response

  1. The novelty of the paper is not justified.

Response:Thank you very much for your comments, reviewer, I have benefited a lot here, I have carefully followed the completion of the comments you gave to complete the appropriate changes.

1、 In Figure 3(b), I added a PCB circuit board and supplemented that the tested prototype PCB circuit board is a two-layer non-copper design. The device is powered by a 5V power bank, and the communication method is through sharing a WiFi hotspot from a mobile phone.

2、the overall design of the device is inexpensive, more than 20 times cheaper than the traditional spectrometer equipment, through the collected data analysis and judgment, its accuracy is also within the acceptable range.

3、The data storage method of this device is different from that of traditional spectrometers. The data from traditional spectrometers can only be stored locally, while the device designed in this paper can be directly transmitted to a cloud-based server database for remote viewing and downloading.

4、We provide a visual web page interface, which allows data collection and viewing operations to be performed on any device that can open a web page. In contrast, most traditional spectrometers can only perform related local operations on Windows, which limits the development of the device.

5、 In Appendix B, we added a data transmission performance test, and the test results showed that the device has a high response speed and no data loss, making it capable of wireless data transmission tasks.

  1. The figure of the paper is not clear and up to the mark.

Response:Thank you very much reviewer for your comments, I have re-edited the charts in the text.

  1. It is desired to discuss the model complexity of the proposed model and the baseline models. This would show their efficiency in terms of runtime and space complexity.

Response:Thank you very much reviewer for your comments, I have added the appropriate discussion in Appendix C.

  1. I suggest the authors give a problem statement or use case in Methodology so as to help readers understand how to make a recommendation based on the dataset.

Response:Thank you very much for your comments. Based on your suggestions, I have added a problem statement in the introduction section and provided detailed instructions for the usage in section 2.4. In sections 3.2 and 3.3, we analyzed RMSEc, Rc2, RMSEp, and Rp2 to clarify the accuracy and reliability of the regression model predictions..

  1. I was somewhat confused about the motivations described in the introduction. I’m not sure if the first paragraph is fully necessary since it frames the problem as reducing detection/response time after a fall in order to get a response; is that the primary goal or is it to detect the fall prior to occurrence and prevent it (at no point do the results compare timing for estimating a pre-fall or fall prior to occurrence)? What are the present timescales for detecting and alerting to a fall and how much matters? For example, if a fall has already happened, does a difference between a 0.2s and 2s alert actually matter when the response time may be substantially greater? This would be useful to know whether the detections actually happen in time.

Response:Thank you very much reviewer for your comments, After re-reading the introduction, I did find that the first paragraph was not entirely necessary, so I have deleted and re-edited it.

  1. Do you anticipate using the generated model as is on new data from people who the model has not been trained on or are you expecting to need to expand the training set and include each individual who uses such a device in the training set?

Response:Thank you very much reviewer for your comments, I randomly divided all the data into two parts in a 7:3 ratio, with 70% used for model training and 30% used for model prediction. The trained model was then applied directly to new data that had not been used for training, in order to predict the corresponding chlorophyll SPAD values.

Round 2

Reviewer 1 Report

After reviewing the revised version, it is evident that the authors have thoroughly addressed all the comments. They have made the paper clearer and more readable. Additionally, they have incorporated valuable information into nearly every section, which significantly enhances the clarity of the design, infrastructure, and experiments conducted.

I do not have any other comments and I think the paper could be published in its current form.

I would like to congratulate the authors for this work.

Reviewer 2 Report

  can be accepted in the current version